# PixelGAN Autoencoders

**Alireza Makhzani, Brendan Frey**
University of Toronto
{makhzani,frey}@psi.toronto.edu

## Abstract

In this paper, we describe the "PixelGAN autoencoder", a generative autoencoder in which the generative path is a convolutional autoregressive neural network on pixels (PixelCNN) that is conditioned on a latent code, and the recognition path uses a generative adversarial network (GAN) to impose a prior distribution on the latent code. We show that different priors result in different decompositions of information between the latent code and the autoregressive decoder. For example, by imposing a Gaussian distribution as the prior, we can achieve a global vs. local decomposition, or by imposing a categorical distribution as the prior, we can disentangle the style and content information of images in an unsupervised fashion. We further show how the PixelGAN autoencoder with a categorical prior can be directly used in semi-supervised settings and achieve competitive semi-supervised classification results on the MNIST, SVHN and NORB datasets.

## 1 Introduction

In recent years, generative models that can be trained via direct back-propagation have enabled remarkable progress in modeling natural images. One of the most successful models is the generative adversarial network (GAN) [1], which employs a two player min-max game. The generative model, $G$, samples the prior $p(\mathbf{z})$ and generates the sample $G(\mathbf{z})$. The discriminator, $D(\mathbf{x})$, is trained to identify whether a point $\mathbf{x}$ is a sample from the data distribution or a sample from the generative model. The generator is trained to maximally confuse the discriminator into believing that generated samples come from the data distribution. The cost function of GAN is

$$\min_G \max_D \mathbb{E}_{\mathbf{x} \sim p_{\text{data}}}[\log D(\mathbf{x})] + \mathbb{E}_{\mathbf{z} \sim p(\mathbf{z})}[\log(1 - D(G(\mathbf{z})))].$$

GANs can be considered within the wider framework of implicit generative models [2, 3, 4]. Implicit distributions can be sampled through their generative path, but their likelihood function is not tractable. Recently, several papers have proposed another application of GAN-style algorithms for approximate inference [2, 3, 4, 5, 6, 7, 8, 9]. These algorithms use implicit distributions to learn posterior approximations that are more expressive than the distributions with tractable densities that are often used in variational inference. For example, adversarial autoencoders [6] use a universal approximator posterior as the implicit posterior distribution and use adversarial training to match the aggregated posterior of the latent code to the prior distribution. Adversarial variational Bayes [3, 7] uses a more general amortized GAN inference framework within a maximum-likelihood learning setting. Another type of GAN inference technique is used in the ALI [8] and BiGAN [9] models, which have been shown to approximate maximum likelihood learning [3]. In these models, both the recognition and generative models are implicit and are jointly learnt by an adversarial training process.

Variational autoencoders (VAE) [10, 11] are another state-of-the-art image modeling technique that use neural networks to parametrize the posterior distribution and pair it with a top-down generative network. Both networks are jointly trained to maximize a variational lower bound on the data log-likelihood. A different framework for learning density models is autoregressive neural networks such

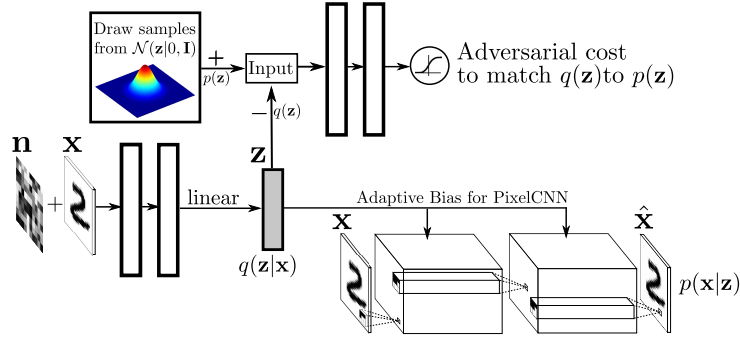

Figure 1: Architecture of the PixelGAN autoencoder.

as NADE [12], MADE [12], PixelRNN [12] and PixelCNN [13]. Unlike variational autoencoders, which capture the statistics of the data in hierarchical latent codes, the autoregressive models learn the image densities directly at the pixel level without learning a hierarchical latent representation.

In this paper, we present the PixelGAN autoencoder as a generative autoencoder that combines the benefits of latent variable models with autoregressive architectures. The PixelGAN autoencoder is a generative autoencoder in which the generative path is a PixelCNN that is conditioned on a latent variable. The latent variable is inferred by matching the aggregated posterior distribution to the prior distribution by an adversarial training technique similar to that of the adversarial autoencoder [6]. However, whereas in adversarial autoencoders the statistics of the data distribution are captured by the latent code, in the PixelGAN autoencoder they are captured jointly by the latent code and the autoregressive decoder. We show that imposing different distributions as the prior results in different factorizations of information between the latent code and the autoregressive decoder. For example, in Section 2.1, we show that by imposing a Gaussian distribution on the latent code, we can achieve a global vs. local decomposition of information. In this case, the global latent code no longer has to model all the irrelevant and fine details of the image, and can use its capacity to capture more relevant and global statistics of the image. Another type of decomposition of information that can be learnt by PixelGAN autoencoders is a discrete vs. continuous decomposition. In Section 2.2, we show that we can achieve this decomposition by imposing a categorical prior on the latent code using adversarial training. In this case, the categorical latent code captures the discrete underlying factors of variation in the data, such as class label information, and the autoregressive decoder captures the remaining continuous structure, such as style information, in an unsupervised fashion. We then show how PixelGAN autoencoders with categorical priors can be directly used in clustering and semi-supervised scenarios and achieve very competitive classification results on several datasets in Section 3. Finally, we present one of the main potential applications of PixelGAN autoencoders in learning cross-domain relations between two different domains in Section 4.

## 2 PixelGAN Autoencoders

Let $\mathbf{x}$ be a datapoint that comes from the distribution $p_{\text{data}}(\mathbf{x})$ and $\mathbf{z}$ be the hidden code. The recognition path of the PixelGAN autoencoder (Figure 1) defines an implicit posterior distribution $q(\mathbf{z}|\mathbf{x})$ by using a deterministic neural function $\mathbf{z} = f(\mathbf{x}, \mathbf{n})$ that takes the input $\mathbf{x}$ along with random noise $\mathbf{n}$ with a fixed distribution $p(\mathbf{n})$ and outputs $\mathbf{z}$. The aggregated posterior $q(\mathbf{z})$ of this model is defined as follows:

$$q(\mathbf{z}) = \int_{\mathbf{x}} q(\mathbf{z}|\mathbf{x}) p_{\text{data}}(\mathbf{x}) d\mathbf{x}.$$

This parametrization of the implicit posterior distribution was originally proposed in the adversarial autoencoder work [6] as the universal approximator posterior. We can sample from this implicit distribution $q(\mathbf{z}|\mathbf{x})$, by evaluating $f(\mathbf{x}, \mathbf{n})$ at different samples of $\mathbf{n}$, but the density function of this posterior distribution is intractable. Appendix A.1 discusses the importance of the input noise in training PixelGAN autoencoders. The generative path $p(\mathbf{x}|\mathbf{z})$ is a conditional PixelCNN [13] that conditions on the latent vector $\mathbf{z}$ using an adaptive bias in PixelCNN layers. The inference is done by an amortized GAN inference technique that was originally proposed in the adversarial autoencoder work [6]. In this method, an adversarial network is attached on top of the hidden code vector of

the autoencoder and matches the aggregated posterior distribution, $q(\mathbf{z})$, to an arbitrary prior, $p(\mathbf{z})$. Samples from $q(\mathbf{z})$ and $p(\mathbf{z})$ are provided to the adversarial network as the negative and positive examples respectively, and the generator of the adversarial network, which is also the encoder of the autoencoder, tries to match $q(\mathbf{z})$ to $p(\mathbf{z})$ by the gradient that comes through the discriminative adversarial network.

The adversarial network, the PixelCNN decoder and the encoder are trained jointly in two phases – the *reconstruction* phase and the *adversarial* phase – executed on each mini-batch. In the reconstruction phase, the ground truth input $\mathbf{x}$ along with the hidden code $\mathbf{z}$ inferred by the encoder are provided to the PixelCNN decoder. The PixelCNN decoder weights are updated to maximize the log-likelihood of the input $\mathbf{x}$. The encoder weights are also updated at this stage by the gradient that comes through the conditioning vector of the PixelCNN. In the adversarial phase, the adversarial network updates both its discriminative network and its generative network (the encoder) to match $q(\mathbf{z})$ to $p(\mathbf{z})$. Once the training is done, we can sample from the model by first sampling $\mathbf{z}$ from the prior distribution $p(\mathbf{z})$, and then sampling from the conditional likelihood $p(\mathbf{x}|\mathbf{z})$ parametrized by the PixelCNN decoder.

We now establish a connection between the PixelGAN autoencoder cost and maximum likelihood learning using a decomposition of the aggregated evidence lower bound (ELBO) proposed in [14]:

$$\mathbb{E}_{\mathbf{x}\sim p_{\text{data}}(\mathbf{x})}[\log p(\mathbf{x})] > -\mathbb{E}_{\mathbf{x}\sim p_{\text{data}}(\mathbf{x})}\Big[\mathbb{E}_{q(\mathbf{z}|\mathbf{x})}[-\log p(\mathbf{x}|\mathbf{z})]\Big] - \mathbb{E}_{\mathbf{x}\sim p_{\text{data}}(\mathbf{x})}\Big[\text{KL}(q(\mathbf{z}|\mathbf{x})\|p(\mathbf{z}))\Big] \quad (1)$$

$$= -\underbrace{\mathbb{E}_{\mathbf{x}\sim p_{\text{data}}(\mathbf{x})}\Big[\mathbb{E}_{q(\mathbf{z}|\mathbf{x})}[-\log p(\mathbf{x}|\mathbf{z})]\Big]}_{\text{reconstruction term}} - \underbrace{\text{KL}(q(\mathbf{z})\|p(\mathbf{z}))}_{\text{marginal KL}} - \underbrace{\mathbb{I}(\mathbf{z};\mathbf{x})}_{\text{mutual info.}} \quad (2)$$

The first term in Equation 2 is the reconstruction term and the second term is the marginal KL divergence between the aggregated posterior and the prior distribution. The third term is the mutual information between the latent code $\mathbf{z}$ and the input $\mathbf{x}$. This is a regularization term that encourages $\mathbf{z}$ and $\mathbf{x}$ to be decoupled by removing the information of the data distribution from the hidden code. If the training set has $N$ examples, $\mathbb{I}(\mathbf{z};\mathbf{x})$ is bounded as follows (see [14]).

$$0 < \mathbb{I}(\mathbf{z};\mathbf{x}) < \log N \quad (3)$$

In order to maximize the ELBO, we need to minimize all the three terms of Equation 2. We consider two cases for the decoder $p(\mathbf{x}|\mathbf{z})$:

**Deterministic Decoder.** If the decoder $p(\mathbf{x}|\mathbf{z})$ is deterministic or has very limited stochasticity such as the simple factorized decoder of the VAE, the mutual information term acts in the complete opposite direction of the reconstruction term. This is because the only way to minimize the reconstruction error of $\mathbf{x}$ is to learn a hidden code $\mathbf{z}$ that is relevant to $\mathbf{x}$, which results in maximizing $\mathbb{I}(\mathbf{z};\mathbf{x})$. Indeed, it can be shown that minimizing the reconstruction term maximizes a variational lower bound on $\mathbb{I}(\mathbf{z};\mathbf{x})$ [15, 16]. For example, in the case of the VAE trained on MNIST, since the reconstruction is precise, the mutual information term is dominated and is close to its maximum value $\mathbb{I}(\mathbf{z};\mathbf{x}) \approx \log N \approx 11.00$ nats [14].

**Stochastic Decoder.** If we use a powerful decoder such as the PixelCNN, the reconstruction term and the mutual information term will not compete with each other anymore and the network can minimize both independently. In this case, the optimal solution for maximizing the ELBO would be to model $p_{\text{data}}(\mathbf{x})$ solely by $p(\mathbf{x}|\mathbf{z})$ and thereby minimizing the reconstruction term, and at the same time, minimizing the mutual information term by ignoring the latent code. As a result, even though the model achieves a high likelihood, the latent code does not learn any useful representation, which is undesirable. This problem has been observed in several previous works [17, 18] and different techniques such as annealing the weight of the KL term [17] or weakening the decoder [18] have been proposed to make $\mathbf{z}$ and $\mathbf{x}$ more dependent.

As suggested in [19, 18], we think that the maximum likelihood objective by itself is not a useful objective for representation learning especially when a powerful decoder is used. In PixelGAN autoencoders, in order to encourage learning more useful representations, we modify the ELBO (Equation 2) by removing the mutual information term from it, since this term is explicitly encouraging $\mathbf{z}$ to become independent of $\mathbf{x}$. So our cost function only includes the reconstruction term and the marginal KL term. The reconstruction term is optimized by the reconstruction phase of training and the marginal KL term is approximately optimized by the adversarial phase[1]. Note that since the

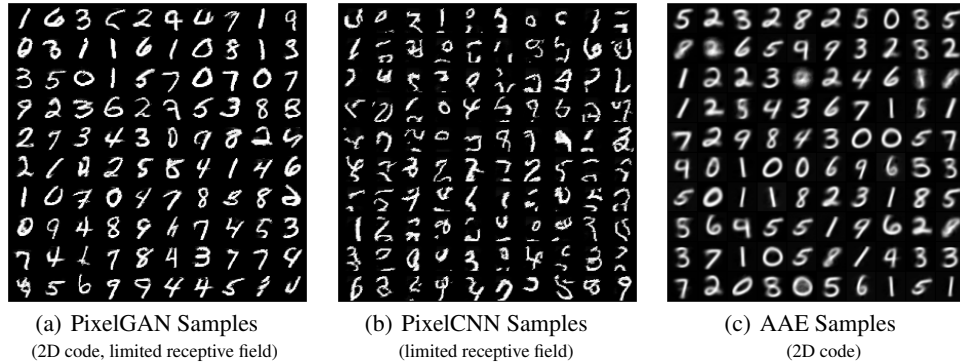

(a) PixelGAN Samples
(2D code, limited receptive field)

(b) PixelCNN Samples
(limited receptive field)

(c) AAE Samples
(2D code)

Figure 2: (a) Samples of the PixelGAN autoencoder with 2D Gaussian code and limited receptive field of size 9. (b) Samples of the PixelCNN (c) Samples of the adversarial autoencoder.

mutual information term is upper bounded by a constant ($\log N$), we are still maximizing a lower bound on the log-likelihood of data. However, this bound is weaker than the ELBO, which is the price that is paid for learning more useful latent representations by balancing the decomposition of information between the latent code and the autoregressive decoder.

For implementing the conditioning adaptive bias in the PixelCNN decoder, we explore two different architectures [13]. In the *location-invariant* bias, for each PixelCNN layer, we use the latent code to construct a vector that is broadcasted within each feature map of the layer and then added as an adaptive bias to that layer. In the *location-dependent* bias, we use the latent code to construct a spatial feature map that is broadcasted across different feature maps and then added only to the first layer of the decoder as an adaptive bias. We will discuss the effect of these architectures on the learnt representation in Figure 3 of Section 2.1 and their implementation details in Appendix A.2.

## 2.1 PixelGAN Autoencoders with Gaussian Priors

Here, we show that PixelGAN autoencoders with Gaussian priors can decompose the global and local statistics of the images between the latent code and the autoregressive decoder. Figure 2a shows the samples of a PixelGAN autoencoder model with the location-dependent bias trained on the MNIST dataset. For the purpose of better illustrating the decomposition of information, we have chosen a 2-D Gaussian latent code and a limited the receptive field of size 9 for the PixelGAN autoencoder. Figure 2b shows the samples of a PixelCNN model with the same limited receptive field size of 9 and Figure 2c shows the samples of an adversarial autoencoder with the 2-D Gaussian latent code. The PixelCNN can successfully capture the local statistics, but fails to capture the global statistics due to the limited receptive field size. In contrast, the adversarial autoencoder, whose sample quality is very similar to that of the VAE, can successfully capture the global statistics, but fails to generate the details of the images. However, the PixelGAN autoencoder, with the same receptive field and code size, can combine the best of both and generates sharp images with global statistics.

In PixelGAN autoencoders, both the PixelCNN depth and the conditioning architecture affect the decomposition of information between the latent code and the autoregressive decoder. We investigate these effects in Figure 3 by training a PixelGAN autoencoder on MNIST where the code size is chosen to be 2 for the visualization purpose. As shown in Figure 3a,b, when a shallow decoder is used, most of the information will be encoded in the hidden code and there is a clean separation between the digit clusters. As we make the PixelCNN more powerful (Figure 3c,d), we can see that the hidden code is still used to capture some relevant information of the input, but the separation of digit clusters is not as sharp when the limited code size of 2 is used. In the next section, we will show that by using a larger code size (e.g., 30), we can get a much better separation of digit clusters even when a powerful PixelCNN is used.

The conditioning architecture also affects the decomposition of information. In the case of the location-invariant bias, the hidden code is encouraged to learn the global information that is location-invariant (the *what* information and not the *where* information) such as the class label information. For example, we can see in Figure 3a,c that the network has learnt to use one of the axes of the 2D Gaussian code to explicitly encode the digit label even though a continuous prior is imposed. In this

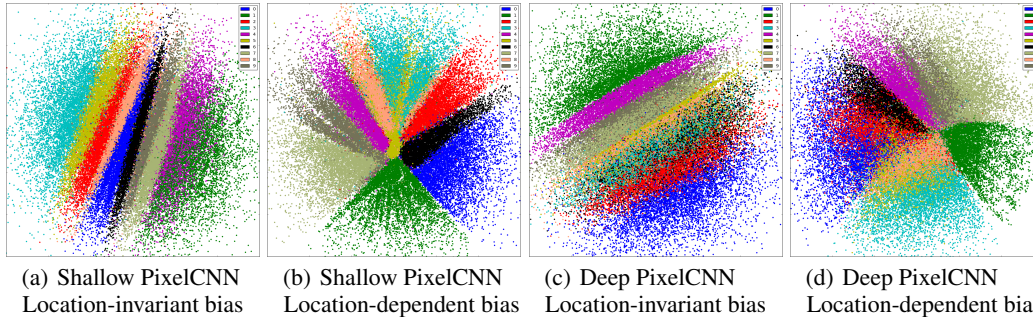

| (a) Shallow PixelCNN | (b) Shallow PixelCNN | (c) Deep PixelCNN | (d) Deep PixelCNN |
| Location-invariant bias | Location-dependent bias | Location-invariant bias | Location-dependent bias |

Figure 3: The effect of the PixelCNN decoder depth and the conditioning architecture on the learnt representation of the PixelGAN autoencoder. (*Shallow*=3 ResBlocks, *Deep*=12 ResBlocks)

case, we can potentially get a much better separation if we impose a discrete prior. This makes this architecture suitable for the discrete vs. continuous decomposition and we use it for our clustering and semi-supervised learning experiments. In the case of the location-dependent bias (Figure 3b,d), the hidden code is encouraged to learn the global information that has location dependent information such as low-frequency content of the image, similar to what the hidden code of an adversarial or variational autoencoder would learn (Figure 2c). This makes this architecture suitable for the global vs. local decomposition experiments such as Figure 2a.

From Figure 3, we can see that the class label information is mostly captured by $p(\mathbf{z})$ while the style information of the images is captured by both $p(\mathbf{z})$ and $p(\mathbf{x}|\mathbf{z})$. This decomposition of information has also been studied in other works that combine the latent variable models with autoregressive decoders such as PixelVAE [20] and variational lossy autoencoders (VLAE) [18]. For example, the VLAE model [18] proposes to use the depth of the PixelCNN decoder to control the decomposition of information. In their model, the PixelCNN decoder is designed to have a shallow depth (small local receptive field) so that the latent code $\mathbf{z}$ is forced to capture more global information. This approach is very similar to our example of the PixelGAN autoencoder in Figure 2. However, the question that has remained unanswered is whether it is possible to achieve a complete decomposition of content and style in an unsupervised fashion, where the class label or discrete structure information is encoded in the latent code $\mathbf{z}$, and the remaining continuous structure such as style is captured by a powerful and deep PixelCNN decoder. This kind of decomposition is particularly interesting as it can be directly used for clustering and semi-supervised classification. In the next section, we show that we can learn this decomposition of content and style by imposing a categorical distribution on the latent representation $\mathbf{z}$ using adversarial training. Note that this discrete vs. continuous decomposition is very different from the global vs. local decomposition, because a continuous factor of variation such as style can have both global and local effect on the image. Indeed, in order to achieve the discrete vs. continuous decomposition, we have to use very deep and powerful PixelCNN decoders (up to 20 residual blocks) to capture both the global and local statistics of the style by the PixelCNN while the discrete content of the image is captured by the categorical latent variable.

## 2.2 PixelGAN Autoencoders with Categorical Priors

In this section, we present an architecture of the PixelGAN autoencoder that can separate the discrete information (e.g., class label) from the continuous information (e.g., style information) in the images. We then show how our architecture can be naturally adopted for the semi-supervised settings.

The architecture that we use is similar to Figure 1, with the difference that we impose a categorical distribution as the prior rather the Gaussian distribution (Figure 4) and also use the location-independent bias architecture. Another difference is that we use a convolutional network as the inference network $q(\mathbf{z}|\mathbf{x})$ to encourage the encoder to preserve the content and lose the style information of the image. The inference network has a softmax output and predicts a one-hot vector whose dimension is the number of discrete labels or categories that we wish the data to be clustered into. The adversarial network is trained directly on the continuous probability outputs of the softmax layer of the encoder.

Imposing a categorical distribution at the output of the encoder imposes two constraints. The first constraint is that the encoder has to make confident decisions about the class labels of the inputs. The

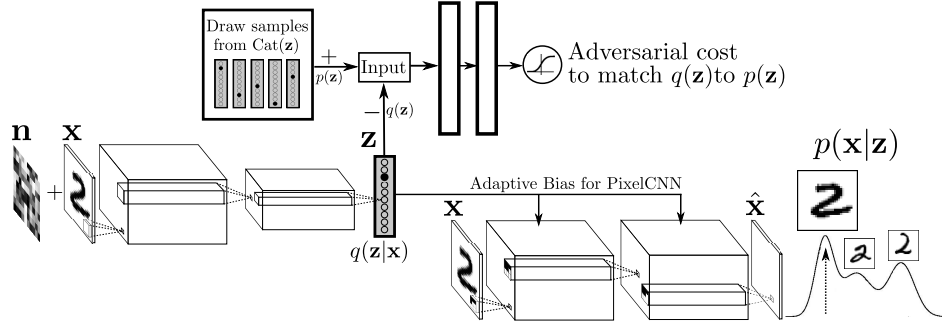

Figure 4: Architecture of the PixelGAN autoencoder with the categorical prior. $p(\mathbf{z})$ captures the class label and $p(\mathbf{x}|\mathbf{z})$ is a multi-modal distribution that captures the style distribution of a digit conditioned on the class label of that digit.

adversarial training pushes the output of the encoder to the corners of the softmax simplex, by which it ensures that the autoencoder cannot use the latent vector $\mathbf{z}$ to carry any continuous style information. The second constraint imposed by adversarial training is that the aggregated posterior distribution of $\mathbf{z}$ should match the categorical prior distribution with uniform outcome probabilities. This constraint enforces the encoder to evenly distribute the class labels across the corners of the softmax simplex. Because of these constraints, the latent variable will only capture the discrete content of the image and all the continuous style information will be captured by the autoregressive decoder.

In order to better understand and visualize the effect of the adversarial training on shaping the hidden code distribution, we train a PixelGAN autoencoder on the first three digits of MNIST (18000 training and 3000 test points) and choose the number of clusters to be 3. Suppose $\mathbf{z} = [z_1, z_2, z_3]$ is the hidden code which in this case is the output probabilities of the softmax layer of the inference network. In Figure 5a, we project the 3D softmax simplex of $z_1 + z_2 + z_3 = 1$ onto a 2D triangle and plot the hidden codes of the training examples when no distribution is imposed on the hidden code. We can see from this figure that the network has learnt to use the surface of the softmax simplex to encode style information of the digits and thus the three corners of the simplex do not have any meaningful interpretation. Figure 5b corresponds to the code space of the same network when a categorical distribution is imposed using the adversarial training. In this case, we can see the network has successfully learnt to encode the label information of the three digits in the three corners of the simplex, and all the style information has been separately captured by the autoregressive decoder. This network achieves an almost perfect test error-rate of $0.3\%$ on the first three digits of MNIST, even though it is trained in a purely unsupervised fashion.

Once the PixelGAN autoencoder is trained, its encoder can be used for clustering new points and its decoder can be used to generate samples from each cluster. Figure 6 illustrates the samples of the PixelGAN autoencoder trained on the full MNIST dataset. The number of clusters is set to be 30 and each row corresponds to the conditional samples of one of the clusters (only 16 are shown). We can see that the discrete latent code of the network has learnt discrete factors of variation such as

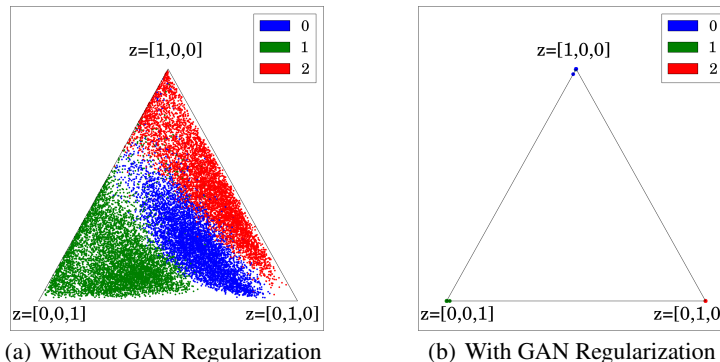

(a) Without GAN Regularization    (b) With GAN Regularization

Figure 5: Effect of GAN regularization (categorical prior) on the code space of PixelGAN autoencoders.

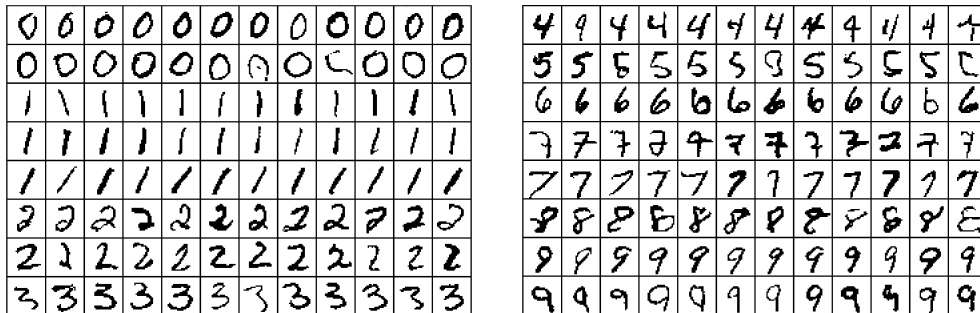

Figure 6: Disentangling the content and style in an unsupervised fashion with PixelGAN autoencoders. Each row shows samples of the model from one of the learnt clusters.

class label information and some discrete style information. For example digit 1s are put in different clusters based on how much tilted they are. The network is also assigning different clusters to digit 2s (based on whether they have a loop) and digit 7s (based on whether they have a dash in the middle). In Section 3, we will show that by using the encoder of this network, we can obtain about 5% error rate in classifying digits in an unsupervised fashion, just by matching each cluster to a digit type.

**Semi-Supervised PixelGAN Autoencoders.** The PixelGAN autoencoder can be used in a semi-supervised setting. In order to incorporate the label information, we add a *semi-supervised* training phase. Specifically, we set the number of clusters to be the same as the number of class labels and after executing the reconstruction and the adversarial phases on an unlabeled mini-batch, the semi-supervised phase is executed on a labeled mini-batch, by updating the weights of the encoder $q(\mathbf{z}|\mathbf{x})$ to minimize the cross-entropy cost. The semi-supervised cost also reduces the mode-missing behavior of the GAN training by enforcing the encoder to learn all the modes of the categorical distribution. In Section 3, we will evaluate the performance of the PixelGAN autoencoders on the semi-supervised classification tasks.

## 3 Experiments

In this paper, we presented the PixelGAN autoencoder as a generative model, but the currently available metrics for evaluating the likelihood of GAN-based generative models such as Parzen window estimate are fundamentally flawed [21]. So in this section, we only present the performance of the PixelGAN autoencoder on downstream tasks such as unsupervised clustering and semi-supervised classification. The details of all the experiments can be found in Appendix B.

**Unsupervised Clustering.** We trained a PixelGAN autoencoder in an unsupervised fashion on the MNIST dataset (Figure 6). We chose the number of clusters to be 30 and used the following evaluation protocol: once the training is done, for each cluster $i$, we found the validation example

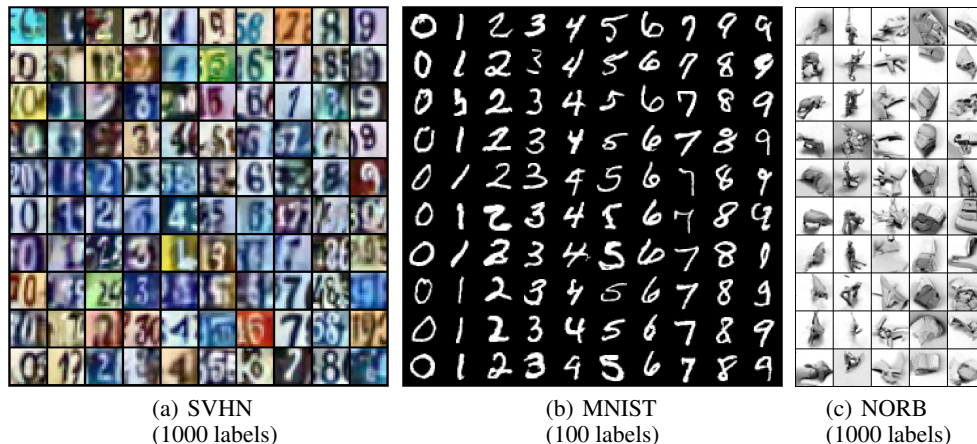

|                (a) SVHN                |                (b) MNIST                |                (c) NORB                |
| :------------------------------------: | :-------------------------------------: | :------------------------------------: |
|             (1000 labels)              |              (100 labels)               |             (1000 labels)              |

Figure 7: Conditional samples of the semi-supervised PixelGAN autoencoder.

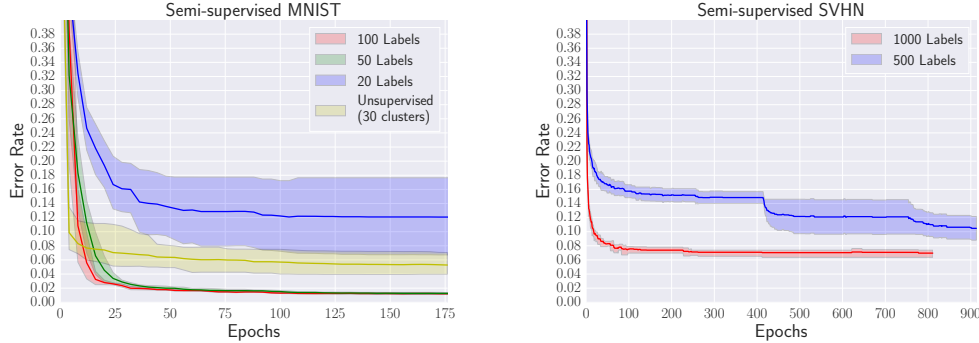

Figure 8: Semi-supervised error-rate of PixelGAN autoencoders on the MNIST and SVHN datasets.

| | MNIST (Unsupervised) | MNIST (20 labels) | MNIST (50 labels) | MNIST (100 labels) | SVHN (500 labels) | SVHN (1000 labels) | NORB (1000 labels) |
|---|---|---|---|---|---|---|---|
| VAE [24] | - | - | - | 3.33 ($\pm0.14$) | - | 36.02 ($\pm0.10$) | 18.79 ($\pm0.05$) |
| VAT [25] | - | - | - | 2.33 | - | 24.63 | 9.88 |
| ADGM [26] | - | - | - | 0.96 ($\pm0.02$) | - | 22.86 | 10.06 ($\pm0.05$) |
| SDGM [26] | - | - | - | 1.32 ($\pm0.07$) | - | 16.61 ($\pm0.24$) | 9.40 ($\pm0.04$) |
| Adversarial Autoencoder [6] | 4.10 ($\pm1.13$) | - | - | 1.90 ($\pm0.10$) | - | 17.70 ($\pm0.30$) | - |
| Ladder Networks [27] | - | - | - | 0.89 ($\pm0.50$) | - | - | - |
| Convolutional CatGAN [22] | 4.27 | - | - | 1.39 ($\pm0.28$) | - | - | - |
| InfoGAN [16] | 5.00 | - | - | - | - | - | - |
| Feature Matching GAN [28] | - | 16.77 ($\pm4.52$) | 2.21 ($\pm1.36$) | 0.93 ($\pm0.06$) | 18.44 ($\pm4.80$) | 8.11 ($\pm1.30$) | - |
| Temporal Ensembling [23] | - | - | - | - | 7.05 ($\pm0.30$) | 5.43 ($\pm0.25$) | - |
| **PixelGAN Autoencoders** | 5.27 ($\pm1.81$) | 12.08 ($\pm5.50$) | 1.16 ($\pm0.17$) | 1.08 ($\pm0.15$) | 10.47 ($\pm1.80$) | 6.96 ($\pm0.55$) | 8.90 ($\pm1.0$) |

Table 1: Semi-supervised learning and clustering error-rate on MNIST, SVHN and NORB datasets.

$x_n$ that maximizes $q(z_i|x_n)$, and assigned the label of $x_n$ to all the points in the cluster $i$. We then computed the test error based on the assigned class labels to each cluster. As shown in the first column of Table 1, the performance of PixelGAN autoencoders is on par with other GAN-based clustering algorithms such as CatGAN [22], InfoGAN [16] and adversarial autoencoders [6].

**Semi-supervised Classification.** Table 1 and Figure 8 report the results of semi-supervised classification experiments on the MNIST, SVHN and NORB datasets. On the MNIST dataset with 20, 50 and 100 labels, our classification results are highly competitive. Note that the classification rate of unsupervised clustering of MNIST is better than semi-supervised MNIST with 20 labels. This is because in the unsupervised case, the number of clusters is 30, but in the semi-supervised case, there are only 10 class labels which makes it more likely to confuse two digits. On the SVHN dataset with 500 and 1000 labels, the PixelGAN autoencoder outperforms all the other methods except the recently proposed temporal ensembling work [23] which is not a generative model. On the NORB dataset with 1000 labels, the PixelGAN autoencoder outperforms all the other reported results.

Figure 7 shows the conditional samples of the semi-supervised PixelGAN autoencoder on the MNIST, SVHN and NORB datasets. Each column of this figure presents sampled images conditioned on a fixed one-hot latent code. We can see from this figure that the PixelGAN autoencoder can achieve a rather clean separation of style and content on these datasets with very few labeled data.

## 4 Learning Cross-Domain Relations with PixelGAN Autoencoders

In this section, we discuss how the PixelGAN autoencoder can be viewed in the context of learning cross-domain relations between two different domains. We also describe how the problem of clustering or semi-supervised learning can be cast as the problem of finding a smooth cross-domain mapping from the data distribution to the categorical distribution.

Recently several GAN-based methods have been developed to learn a cross-domain mapping between two different domains [29, 30, 31, 6, 32]. In [31], an unsupervised cost function called the output distribution matching (ODM) is proposed to find a cross-domain mapping $F$ between two domains $\mathbb{D}_1$ and $\mathbb{D}_2$ by imposing the following unsupervised constraint on the uncorrelated samples from $\mathbf{x} \sim \mathbb{D}_1$ and $\mathbf{y} \sim \mathbb{D}_2$:

$$\text{Distr}[F(\mathbf{x})] = \text{Distr}[\mathbf{y}] \tag{4}$$

where Distr[$\mathbf{z}$] denotes the distribution of the random variable $\mathbf{z}$. The adversarial training is proposed as one of the methods for matching these distributions. If we have access to a few labeled pairs $(\mathbf{x}, \mathbf{y})$, then $F$ can be further trained on them in a supervised fashion to satisfy $F(\mathbf{x}) = \mathbf{y}$. For example, in speech recognition, we want to find a cross-domain mapping from a sequence of phonemes to a sequence of characters. By optimizing the ODM cost function in Equation 4, we can find a smooth function $F$ that takes phonemes at its input and outputs a sequence of characters that respects the language model. However, the main problem with this method is that the network can learn to ignore part of the input distribution and still satisfy the ODM cost function by its output distribution. This problem has also been observed in other works such as [29]. One way to avoid this problem is to add a reconstruction term to the ODM cost function by introducing a reverse mapping from the output of the encoder to the input domain. The is essentially the idea of the adversarial autoencoder (AAE) [6] which learns a generative model by finding a cross-domain mapping between a Gaussian distribution and the data distribution. Using the ODM cost function along with a reconstruction term to learn cross-domain relations have been explored in several previous works. For example, InfoGAN [16] adds a mutual information term to the ODM cost function and optimizes a variational lower bound on this term. It can be shown that maximizing this variational bound is indeed minimizing the reconstruction cost of an autoencoder [15]. Similarly, in [32, 33], an AAE is used to learn the cross-domain relations of the vector representations of words from two different languages. The architecture of the recent works of DiscoGAN [29] and CycleGAN [30] are also similar to an AAE in which the latent representation is enforced to have the distribution of the other domain. Here we describe how our proposed PixelGAN autoencoder can be potentially used in all these application areas to learn better cross-domain relations. Suppose we want to learn a mapping from domain $\mathbb{D}_1$ to $\mathbb{D}_2$. In the architecture of Figure 1, we can use independent samples of $\mathbf{x} \sim \mathbb{D}_1$ at the input and instead of imposing a Gaussian distribution on the latent code, we can impose the distribution of the second domain using its independent samples $\mathbf{y} \sim \mathbb{D}_2$. Unlike AAEs, the encoder of PixelGAN autoencoders does not have to retain all the input information in order to have a lossless reconstruction. So the encoder can use all its capacity to learn the most relevant mapping from $\mathbb{D}_1$ to $\mathbb{D}_2$ and at the same time, the PixelCNN can capture the remaining information that has been lost by the encoder.

We can adopt the ODM idea for semi-supervised learning by assuming $\mathbb{D}_1$ is the image domain and $\mathbb{D}_2$ is the label domain. Independent samples of $\mathbb{D}_1$ and $\mathbb{D}_2$ correspond to samples from the data distribution $p_{\text{data}}(\mathbf{x})$ and the categorical distribution. The function $F = q(\mathbf{y}|\mathbf{x})$ can be parametrized by a neural network that is trained to satisfy the ODM cost function by matching the aggregated distribution $q(\mathbf{y}) = \int q(\mathbf{y}|\mathbf{x})p_{\text{data}}(\mathbf{x})d\mathbf{x}$ to the categorical distribution using adversarial training. The few labeled examples are used to further train $F$ to satisfy $F(\mathbf{x}) = \mathbf{y}$. However, as explained above, the problem with this method is that the network can learn to generate the categorical distribution by ignoring some part of the input distribution. The AAE solves this problem by adding an inverse mapping from the categorical distribution to the data distribution. However, the main drawback of the AAE architecture is that due to the reconstruction term, the latent representation now has to model all the underlying factors of variation in the image. For example, in the semi-supervised AAE architecture [6], while we are only interested in the one-hot label representation to do semi-supervised learning, we also need to infer the style of the image so that we can have a lossless reconstruction of the image. The PixelGAN autoencoder solves this problem by enabling the encoder to only infer the factor of variation that we are interested in (i.e., label information), while the remaining structure of the input (i.e., style information) is automatically captured by the autoregressive decoder.

## 5   Conclusion

In this paper, we proposed the PixelGAN autoencoder, which is a generative autoencoder that combines a generative PixelCNN with a GAN inference network that can impose arbitrary priors on the latent code. We showed that imposing different distributions as the prior enables us to learn a latent representation that captures the type of statistics that we care about, while the remaining structure of the image is captured by the PixelCNN decoder. Specifically, by imposing a Gaussian prior, we were able to disentangle the low-frequency and high-frequency statistics of the images, and by imposing a categorical prior we were able to disentangle the style and content of images and learn representations that are specifically useful for clustering and semi-supervised learning tasks. While the main focus of this paper was to demonstrate the application of PixelGAN autoencoders in downstream tasks such as semi-supervised learning, we discussed how these architectures have many other potentials such as learning cross-domain relations between two different domains.

## Acknowledgments

We would like to thank Nathan Killoran for helpful discussions. We also thank NVIDIA for GPU donations.

## Footnotes

[1]The original GAN formulation optimizes the Jensen-Shannon divergence [1], but there are other formulations that optimize the KL divergence, e.g. [3].

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
