[Supplementary Material]

# Appendix A  Implementation Details

In this section, we describe two important architecture design choices for training PixelGAN autoencoders.

## A.1  Input noise

In all the semi-supervised experiments, we found it crucial to use the universal approximator posterior discussed in Section 2, as opposed to a deterministic posterior. Specifically, the input noise that we use is an additive Gaussian noise, which results in a posterior distribution $q(\mathbf{z}|\mathbf{x})$ that is more expressive than that of a model without the input corruption. This is similar to the denoising criterion idea proposed in [34]. We believe this additive noise is also playing an important role in preventing the mode-missing behavior of the GAN when imposing a degenerate distribution such as the categorical distribution. Similar related ideas have been used to stabilize GAN training such as instance noise [35] or one-sided label noise [28].

## A.2  Conditioning of PixelCNN

There are three methods to implement how the PixelCNN conditions on the latent vector.

**Location-Invariant Bias.**  This is the method that was proposed in the conditional PixelCNN model [13]. Suppose the size of the convolutional layer of the decoder is (`batch, width, height, channels`). Then the PixelCNN can use a linear mapping to convert the conditioning tensor of size (`batch, condition_size`) to generate a tensor of size (`batch, channels`) that is then broadcasted and added to the feature maps of all the layers of the PixelCNN decoder as an adaptive bias. In this method, the hidden code is encouraged to learn the global information that is location-invariant (the *what* information and not the *where* information) such as the class label information. We use this method in all the clustering and semi-supervised learning experiments.

**Location-Dependent Bias.**  Suppose the size of the convolutional layer of the PixelCNN decoder is (`batch, width, height, channels`). Then the PixelCNN can use a one layer neural network to convert the conditioning tensor of size (`batch, condition_size`) to generate a spatial tensor of size (`batch, width, height, k`) followed by a $1 \times 1$ convolutional layer to construct a tensor of size (`batch, width, height, channels`) that is then added only to the feature maps of the first layer of the decoder as an adaptive bias (similar to the VPN model [36]). When $k = 1$, we can simply broadcast the tensor of size (`batch, width, height, k=1`) to get a tensor of size (`batch, width, height, channels`) instead of using the $1 \times 1$ convolution. In this method, the latent vector has spatial and location-dependent information within the feature map. This is the method that we used in experiments of Figure 2a.

**Input Channel.**  Another method for conditioning is proposed in the PixelVAE [20] and the variational lossy autoencoder (VLAE) [18]. In this method, first a tensor of size (`batch, width, height, k`) is constructed using the conditioning tensor similar to the location-dependent bias. This tensor is then concatenated to the input of the PixelCNN. The performance and computational complexity of this method is very similar to that of the location-dependent bias method.

# Appendix B  Experiment Details

We used TensorFlow [37] in all of our experiments.

## B.1  MNIST Dataset

The MNIST dataset has 50K training points, 10K validation points and 10K test points. We perform experiments on both the binary MNIST and the real-valued MNIST. In the real valued MNIST experiments, we subtract 127.5 from the data points and then divide them by 127.5 and use the discretized logistic mixture likelihood [38] as the cost function for the PixelCNN. In the case of binary MNIST, the data points are binarized by setting pixel values larger than 0.5 to 1, and values smaller than 0.5 to 0.

### B.1.1 PixelGAN Autoencoders with Gaussian Prior on MNIST

Here we describe the model architecture used for training the PixelGAN autoencoder with a Gaussian prior on the binary MNIST dataset in Figure 2a. The PixelCNN decoder uses both the vertical and horizontal stacks similar to [13]. The cost function of the PixelCNN is the cross-entropy cost function. The PixelCNN uses the location-dependent bias as described in Appendix A.2. Specifically, a tensor of size (`batch`, `width`, `height`, `1`) is constructed from the conditioning vector by using a one-layer neural network with 1000 hidden units, ReLU activation and linear output. This tensor is then broadcasted and added only to the feature maps of the first layer of the PixelCNN decoder. The PixelCNN is designed to have a local receptive field by having 3 residual blocks (filter size of 3x5, 32 feature maps, ReLU non-linearity as in [13]). The adversarial discriminator has two layers of 2000 hidden units with ReLU activation function. The encoder architecture has two fully-connected layers of size 2000 with ReLU non-linearity. The last layer of the encoder $q(\mathbf{z}|\mathbf{x})$ has a linear activation function. On the latent representation of size 2, we impose a Gaussian distribution with standard deviation of 5. We used the gradient descent with momentum algorithm for optimizing all the cost functions of the network. For the PixelCNN reconstruction cost, we used the learning rate of 0.001 and the momentum value of 0.9. After 25 epochs we reduce the learning rate to 0.0001. For both of the generator and the discriminator costs, the learning rates and the momentum values were set to 0.1.

### B.1.2 Unsupervised Clustering of MNIST

Here we describe the model architecture used for clustering the binary MNIST dataset in Figure 6 and Section 3. The PixelCNN decoder uses both the vertical and horizontal stacks similar to [13]. The cost function of the PixelCNN is the cross-entropy cost function. The PixelCNN uses the location-invariant bias as described in Appendix A.2 and has 15 residual blocks (filter size of 3x5, 32 feature maps, ReLU non-linearity as in [13]). The adversarial discriminator has two layers of 3000 hidden units with ReLU activation function. The encoder architecture has a convolutional layer (filter size of 7, 32 feature maps, ReLU activation) and a max-pooling layer (pooling size 2), followed by another convolutional layer (filter size of 7, 32 feature maps, ReLU activation) and a max-pooling layer (pooling size 2) with no fully-connected layer. The last layer of the encoder $q(\mathbf{z}|\mathbf{x})$ has the softmax activation function. We found it important to use batch-normalization [39] for all the layers of the encoder *including* the softmax layer. The number of clusters is chosen to be 30. The clusters are represented by a discrete one-hot variable of size 30. On the continuous probability output of the softmax, we impose a categorical distribution with uniform probabilities. We use Adam [40] optimizer with learning rate of 0.001 for optimizing the PixelCNN reconstruction cost function, but we found it important to use the gradient descent with momentum algorithm for optimizing the generator and the discriminator costs of the adversarial network. For both of the generator and the discriminator costs, the momentum values were set to 0.1 and the learning rates were set to 0.01. We use an input dropout noise with the keep probability of 0.8 at the input layer and only at the training time.

The model architecture used for Figure 5 is the same as this architecture except that the number of clusters is chosen to be 3.

### B.1.3 Semi-Supervised MNIST

We performed semi-supervised learning experiments on both binary and real-valued MNIST dataset. We found that the semi-supervised error-rate of the real-valued MNIST is roughly the same as the binary MNIST (about 1.10% with 100 labels), but it takes longer to train due to the logistic mixture likelihood cost function [38]. So in Table 1, we only report the performance with the binary MNIST, but in Figure 7b we are showing the samples of the real-valued MNIST with 100 labels.

**Binary MNIST.** Here we describe the model architecture used for the semi-supervised learning experiments on the binary MNIST in Section 3 and Table 1. The PixelCNN decoder uses both the vertical and horizontal stacks similar to [13] and uses the cross-entropy cost function. The PixelCNN uses the location-invariant bias as described in Appendix A.2. The PixelCNN has 6 residual blocks (filter size of 3x5, 32 feature maps, ReLU non-linearity as in [13]). The adversarial discriminator has two layers of 1000 hidden units with ReLU activation function. The encoder architecture has three convolutional layers (filter size of 5, 32 feature maps, ReLU activation) and a max-pooling layer (pooling size 2), followed by another three convolutional layers (filter size of 5, 32 feature maps, ReLU activation) and a max-pooling layer (pooling size 2) with no fully-connected layer. The

last layer of the encoder $q(\mathbf{z}|\mathbf{x})$ has the softmax activation function. All the convolutional layers of the encoder except the softmax layer use batch-normalization [39]. On the latent representation, we impose a categorical distribution with uniform probabilities. The semi-supervised cost is the cross-entropy cost function at the output of $q(\mathbf{z}|\mathbf{x})$. We use Adam [40] optimizer with learning rate of 0.001 for optimizing the PixelCNN cost and the cross-entropy cost, but we found it important to use the gradient descent with momentum algorithm for optimizing the generator and the discriminator costs of the adversarial network. For both of the generator and the discriminator costs, the momentum values were set to 0.1 and the learning rates were set to 0.1. We add a Gaussian noise with standard deviation of 0.3 to the input layer as described in Appendix A.1. The labeled examples were chosen at random but evenly distributed across the classes.

**Real-valued MNIST.** Here we describe the model architecture used for the semi-supervised learning experiments on the real-valued MNIST in Figure 7b. The PixelCNN decoder uses both the vertical and horizontal stacks similar to [13] and uses a discretized logistic mixture likelihood cost function with 10 logistic distribution as proposed in [38]. The PixelCNN uses the location-invariant bias as described in Appendix A.2. The PixelCNN has 20 residual blocks (filter size of 2x3, 64 feature maps, gated `sigmoid-tanh` non-linearity as in [13]). The adversarial discriminator has two layers of 1000 hidden units with `ReLU` activation function. The encoder architecture has three convolutional layers (filter size of 5, 32 feature maps, `ReLU` activation) and a max-pooling layer (pooling size 2), followed by another three convolutional layers (filter size of 5, 32 feature maps, `ReLU` activation) and a max-pooling layer (pooling size 2) with no fully-connected layer. The last layer of the encoder $q(\mathbf{z}|\mathbf{x})$ has the softmax activation function. All the convolutional layers of the encoder except the softmax layer use batch-normalization [39]. On the latent representation, we impose a categorical distribution with uniform probabilities. The semi-supervised cost is the cross-entropy cost function at the output of $q(\mathbf{z}|\mathbf{x})$. We use Adam [40] optimizer with learning rate of 0.001 for optimizing the PixelCNN cost and the cross-entropy cost, but we found it important to use the gradient descent with momentum algorithm for optimizing the generator and the discriminator costs of the adversarial network. For both of the generator and the discriminator costs, the momentum values were set to 0.1 and the learning rates were set to 0.1. After 150 epochs, we divide all the learning rates by 10. We add a Gaussian noise with standard deviation of 0.3 to the input layer as described in Appendix A.1. The labeled examples were chosen at random but evenly distributed across the classes.

## B.2 SVHN Dataset

The SVHN dataset has about 530K training points and 26K test points. We use 10K points for the validation set. Similar to [25], we downsample the images from $32 \times 32 \times 3$ to $16 \times 16 \times 3$ and then subtracte 127.5 from the data points and then divide them by 127.5.

### B.2.1 Semi-Supervised SVHN

Here we describe the model architecture used for the semi-supervised learning experiments on the SVHN dataset in Section 3. The PixelCNN decoder uses both the vertical and horizontal stacks similar to [13]. The cost function of the PixelCNN is a discretized logistic mixture likelihood cost function with 10 logistic distribution as proposed in [38]. The PixelCNN uses the location-invariant bias as described in Appendix A.2 and has 20 residual blocks (filter size of 3x5, 32 feature maps, gated `sigmoid-tanh` non-linearity as in [13]). The adversarial discriminator has two layers of 1000 hidden units with `ReLU` activation function. The encoder architecture has two convolutional layers (filter size of 5, 32 feature maps, `ReLU` activation) and a max-pooling layer (pooling size 2), followed by another two convolutional layers (filter size of 5, 32 feature maps, `ReLU` activation) and a max-pooling layer (pooling size 2) with no fully-connected layer. The last layer of the encoder $q(\mathbf{z}|\mathbf{x})$ has the softmax activation function. All the convolutional layers of the encoder except the softmax layer use batch-normalization [39]. On the latent representation, we impose a categorical distribution with uniform probabilities. The semi-supervised cost is the cross-entropy cost function at the output of $q(\mathbf{z}|\mathbf{x})$. We use Adam [40] optimizer for optimizing all the cost function. For the PixelCNN cost and the cross-entropy cost we use the learning rate of 0.001 and for the generator and the discriminator costs of the adversarial network we use the learning rate of 0.0001. We add a Gaussian noise with standard deviation of 0.2 to the input layer as described in Appendix A.1.

## B.3  NORB Dataset

The NORB dataset has about 24K training points and 24K test points. We use 4K points for the validation set. This dataset has 5 object categories: animals, human figures, airplanes, trucks and cars. We downsample the images to have the size of $32 \times 32 \times 1$, subtract 127.5 from the data points and then divide them by 127.5.

### B.3.1  Semi-Supervised NORB

The PixelCNN decoder uses both the vertical and horizontal stacks similar to [13]. The cost function of the PixelCNN is a discretized logistic mixture likelihood cost function with 10 logistic distribution as proposed in [38]. The PixelCNN uses the location-invariant bias as described in Appendix A.2 and has 15 residual blocks (filter size of 3x5, 32 feature maps, gated `sigmoid-tanh` non-linearity as in [13]). The adversarial discriminator has two layers of 1000 hidden units with `ReLU` activation function. The encoder architecture has a convolutional layer (filter size of 7, 32 feature maps, `ReLU` activation) and a max-pooling layer (pooling size 2), followed by another convolutional layer (filter size of 7, 32 feature maps, `ReLU` activation) and a max-pooling layer (pooling size 2), followed by another convolutional layer (filter size of 7, 32 feature maps, `ReLU` activation) and a max-pooling layer (pooling size 2) with no fully-connected layer. The last layer of the encoder $q(\mathbf{z}|\mathbf{x})$ has the softmax activation function. All the convolutional layers of the encoder except the softmax layer use batch-normalization [39]. On the latent representation, we impose a categorical distribution with uniform probabilities. The semi-supervised cost is the cross-entropy cost function at the output of $q(\mathbf{z}|\mathbf{x})$. We use Adam [40] optimizer for optimizing all the cost function. For the PixelCNN cost and the cross-entropy cost we use the learning rate of $0.001$ and for the generator and the discriminator costs of the adversarial network we use the learning rate of $0.0001$. We add a Gaussian noise with standard deviation of 0.3 to the input layer as described in Appendix A.1. The labeled examples were chosen at random but evenly distributed across the classes. In the case of NORB with 1000 labels, the test error after 10 epochs is 12.97%, after 100 epochs is 11.63% and after 500 epochs is 8.17%.