[Reviews · NeurIPS 2017]

Reviewer 1



Update after rebuttal: I believe the paper can be accepted as a poster. I advise the authors to polish the writing to better highlight their contributions, motivation and design choices. This could make the work attractive and rememberable, not "yet another hybrid generative model". ----- The paper proposes PixelGAN autocencoder - a generative model which is a hybrid of an adversarial autoencoder and a PixelCNN autoencoder. The authors provide a theoretical justification of the approach based on a decomposition of variational evidence lower bound (ELBO). The authors provide qualitative results with different priors on the hidden distribution, and quantitative results on semi-supervised learning on MNIST, SVHN and NORB. The paper is closely related to Adversarial autoencoders (Makhzani et al. ICLR 2016 Workshop), which, as far as I know, have only been published on arxiv and as a Workshop contribution to ICLR. Yet, the work is well known and widely cited. The novelty of the present submission significantly depends on Adversarial autoencoders being considered existing approach or not. This is a complicated situation, which, I assume, ACs and PCs are better qualified to judge about. Now to some more detailed comments. Pros: 1) Good results on semi-supervised learning on MNIST, SVHN and NORB, and unsupervised clustering on MNIST. It is difficult to say if the results are state-of-the-art since many numbers for the baselines are missing, but at least they are close to the state of the art. 2) A clear discussion of an ELBO decomposition and related architectural choices. Cons: 1) If Adversarial autoencoders are considered existing work, then novelty is somewhat limited - it’s yet another paper which combines two existing generative models. What makes exactly this combination especially interesting? 2) Lack of results on actual image generation. I understand that image generation is difficult to evaluate, but still some likelihood bounds (are these even computable?), Inception scores and images would be nice to see, at least in the Appendix. 3) It is somewhat confusing that two versions of the approach - with location-dependent and location-independent biases - are used in experiments interchangeably, but are not directly compared to each other. I appreciate the authors mentioning this in lines 153-157, but a more in-depth analysis would be useful. 4) A more detailed discussion of relation to existing approaches, such as VLAE and PixelVAE (both published at ICLR 2017), would be helpful. Lines 158-163 are helpful, but do not quite clear highlight the differences, and strengths and weaknesses of different approaches. 5) Some formulations are not quite clear, such as “limited stochasticity” vs “powerful decoder” in lines 88 and 96. Also the statement in line 111 about “approximately optimizing the KL divergence” and the corresponding footnote looks a bit too abstract - so do the authors optimize it or not? 6) In the bibliography the authors tend to ignore the ICLR conference and list many officially published papers as arxiv. 7) Putting a whole section on cross-domain relations to the appendix is not good practice at all. I realize it’s difficult to fit all content to 8 pages, but it’s the job of the authors to organize the paper in such a way that all important contributions fit into the main paper. Overall, I am in the borderline mode. The results are quite good, but the novelty seems limited.

Reviewer 2



The paper build and auto-encoder with pixelCNN decoder and adversarial cost on latent between uniform prior and inference distribution. With the right network design the networks separate global input information stored in the latent and local one captured by pixelCNN when trained on MNIST dataset. With categorical distribution of latents the network learns to capture very close to class information in unsupervised way. The networks perform well in semisupervised settings. The paper is yet another combination of VAE/AdvNet/PixelCNN. The paper has a nice set of experiments and discussion. The model is most closely related to VAE-pixelCNN combination with VAE loss (KL) on latents replaced by adversarial loss (even though they discuss the mathematical difference) and it would be good to run the same experiments (scaling latent loss) with that and compare. More details: - In the Figure 2c I would rather see VAE-pixelCNN combination with the same networks and different scaling of KL term. While there is a mathematical description of the difference, in the end both are some terms penalizing latents, both wanting latents to be unit gaussians. - Line 98: That’s not necessarily the case, depending on function approximations. The network can decide to put information into latents or keep them in the input. But yes, it is harder for it be in the latents since latents are noisy and the posterior is approximate. - Do you get any clear unsupervised separation as in Figure 6 for SVHN and NORB?

Reviewer 3



The paper describes an adversarial autoencoder model (i.e., an autoencoder where the latents are encouraged to match a prior distribution through a GAN-like training scheme), where the decoder is autoregressive. This model can use both its latent representation and the autoregressive connections to model the data, and which part of the model learns what can be manipulated by slightly modifying its architecture and the prior distribution used for the latents. The experiment showcased in Figure 2 is great, it nicely demonstrates the benefits of combining both autoregressive and latent variable based modelling. It's also cool to see that the model learns to encode the digit label despite the continuity of the prior (Figure 3). One thing I'm not sure about is how much the different decompositions (global - local structure vs digit identity - details) are due to the different choice of priors. This is not the only thing that changes between the two types of models. The model with a categorical prior also has a different biasing mechanism, and is much deeper (end of Section 2.1). So I think the paper is a bit quick to attribute the learning of a different decomposition entirely to the change of prior. It would be interesting to take the model with the categorical prior (and all other modifications), and change only the prior back to Gaussian. My guess is it would still learn largely the same decomposition (as in Figure 3(a)), because by my intuition, the architectural changes are much more influential in this than the choice of prior. It would be great to see this addressed in the paper. The experiment in Figure 5 is also very nice as it clearly demonstrates the "discretizing" effect of the GAN loss on the latent code, and the semi-supervised classification experiment is also a great addition. Overall, the model is a natural extension of both adversarial autoencoders and autoregressive models, and the paper presents a nice follow-up to VLAE and PixelVAE (which study VAEs with autoregressive decoders). The experimental section is well thought-out and the results are convincing, although as stated before I'm not convinced that the choice of prior deserves as much credit as it gets for the different decompositions that are learned. Remarks: - Line 45: "the global latent code no longer has to model all the irrelevant and fine details", whether this is useful is very task-dependent, imagine a texture classification task for example. So this statement is a bit overgeneralised. - L187-188: the adversarial net gets continuous inputs (probabilities) from the softmax layer. What about the decoder? I'm guessing this gets discrete input, but this should probably be mentioned explicitly for clarity's sake. - Although scaling up is clearly not the focus of the paper, an experiment on a larger dataset would also be nice. My intuition is that this model will scale beyond regular PixelCNNs because it can effectively use the latents to code for global structure.